# Limited memory Kelley's Method Converges for Composite Convex and Submodular Objectives

**Song Zhou**
Cornell University
sz557@cornell.edu

**Swati Gupta**
Georgia Institute of Technology
swatig@gatech.edu

**Madeleine Udell**
Cornell University
udell@cornell.edu

## Abstract

The original simplicial method (OSM), a variant of the classic Kelley's cutting plane method, has been shown to converge to the minimizer of a composite convex and submodular objective, though no rate of convergence for this method was known. Moreover, OSM is required to solve subproblems in each iteration whose size grows linearly in the number of iterations. We propose a limited memory version of Kelley's method (L-KM) and of OSM that requires limited memory (at most $n + 1$ constraints for an $n$-dimensional problem) independent of the iteration. We prove convergence for L-KM when the convex part of the objective ($g$) is strongly convex and show it converges linearly when $g$ is also smooth. Our analysis relies on duality between minimization of the composite objective and minimization of a convex function over the corresponding submodular base polytope. We introduce a limited memory version, L-FCFW, of the Fully-Corrective Frank-Wolfe (FCFW) method with approximate correction, to solve the dual problem. We show that L-FCFW and L-KM are dual algorithms that produce the same sequence of iterates; hence both converge linearly (when $g$ is smooth and strongly convex) and with limited memory. We propose L-KM to minimize composite convex and submodular objectives; however, our results on L-FCFW hold for general polytopes and may be of independent interest.

## 1 Introduction

One of the earliest and fundamental methods to minimize non-smooth convex objectives is Kelley's method, which minimizes the maximum of lower bounds on the convex function given by the supporting hyperplanes to the function at each previously queried point. An approximate solution to the minimization problem is found by minimizing this piecewise linear approximation, and the approximation is then strengthened by adding the supporting hyperplane at the current approximate solution [11, 6]. Many variants of Kelley's method have been analyzed in the literature [16, 12, 8, for e.g.]. Kelley's method and its variants are a natural fit for problem involving a piecewise linear function, such as composite convex and submodular objectives. This paper defines a new limited memory version of Kelley's method adapted to composite convex and submodular objectives, and establishes the first convergence rate for such a method, solving the open problem proposed in [2, 3].

Submodularity is a discrete analogue of convexity and has been used to model combinatorial constraints in a wide variety of machine learning applications, such as MAP inference, document summarization, sensor placement, clustering, image segmentation [3, and references therein]. Submodular set functions are defined with respect to a ground set of elements $V$, which we may identify with $\{1, \ldots, n\}$ where $|V| = n$. These functions capture the property of diminishing returns: $F : \{0, 1\}^n \to \mathbb{R}$ is said to be submodular if $F(A \cup \{e\}) - F(A) \geq F(B \cup \{e\}) - F(B)$ for all $A \subseteq B \subseteq V$, $e \notin B$. Lovász gave a convex extension $f : [0, 1]^n \to \mathbb{R}$ of the submodular set functions which takes the value of the set function on the vertices of the $[0, 1]^n$ hypercube, i.e. $f(\mathbf{1}_S) = F(S)$, where $\mathbf{1}_S$ is the indicator vector of the set $S \subseteq V$ [17]. (See Section 2 for details.)

In this work, we propose a variant of Kelley's method, LIMITED MEMORY KELLEY'S METHOD (L-KM), to minimize the composite convex and submodular objective

$$\text{minimize} \quad g(x) + f(x), \tag{P}$$

where $g : \mathbb{R}^n \to \mathbb{R}$ is a closed strongly convex proper function and $f : \mathbb{R}^n \to \mathbb{R}$ is the Lovász extension (see Section 2 for details) of a given submodular function $F : 2^{|E|} \to \mathbb{R}$. Composite convex and submodular objectives have been used extensively in sparse learning, where the support of the model must satisfy certain combinatorial constraints. L-KM builds on the ORIGINAL SIMPLICIAL METHOD (OSM), proposed by Bach [3] to minimize such composite objectives. At the $i$th iteration, OSM approximates the Lovász extension by a piecewise linear function $f_{(i)}$ whose epigraph is the maximum of the supporting hyperplanes to the function at each previously queried point. It is natural to approximate the submodular part of the objective by a piecewise linear function, since the Lovász extension is piecewise linear (with possibly an exponential number of pieces). OSM terminates once the algorithm reaches the optimal solution, in contrast to a subgradient method, which might continue to oscillate. Contrast OSM with Kelley's Method: Kelley's Method approximates the full objective function using a piecewise linear function, while OSM only approximates the Lovász extension $f$ and uses the exact form of $g$. In [3], the authors show that OSM converges to the optimum; however, no rate of convergence is given. Moreover, OSM maintains an approximation of the Lovász extension by maintaining a set of linear constraints whose size grows linearly with the number of iterations. Hence the subproblems are harder to solve with each iteration.

This paper introduces L-KM, a variant of OSM that uses no more than $n + 1$ linear constraints in each approximation $f_{(i)}$ (and often, fewer) and provably converges when $g$ is strongly convex. When in addition $g$ is smooth, our new analysis of L-KM shows that it converges linearly, and, as a corollary, that OSM also converges linearly, which was previously unknown.

To establish this result, we introduce the algorithm L-FCFW to solve a problem dual to ($P$):

$$\begin{array}{ll} \text{maximize} & h(w) \\ \text{subject to} & w \in B(F), \end{array} \tag{D}$$

where $h : \mathbb{R}^n \to \mathbb{R}$ is a smooth concave function and $B(F) \subset \mathbb{R}^n$ is the submodular base polytope corresponding to a given submodular function $F$ (defined below). We show L-FCFW is a limited memory version of the FULLY-CORRECTIVE FRANK-WOLFE (FCFW) method with approximate correction [15], and hence converges linearly to a solution of ($D$).

We show that L-KM and L-FCFW are *dual algorithms* in the sense that both algorithms produce the same sequence of primal iterates and lower and upper bounds on the objective. This connection immediately implies that L-KM converges linearly. Furthermore, when $g$ is smooth as well as strongly convex, we can recover the dual iterates of L-FCFW from the primal iterates of L-KM.

**Related Work:** The Original Simplicial Method was proposed by Bach (2013) [3]. As mentioned earlier, it converges finitely but no known rate of convergence was known before the present work. In 2015, Lacoste-Julien and Jaggi proved global linear convergence of variants of the Frank-Wolfe algorithm, including the Fully Corrective Frank-Wolfe (FCFW) with approximate correction [15]. L-FCFW, proposed in this paper, can be shown to be a limited memory special case of the latter, which proves linear convergence of both L-KM and OSM.

Many authors have studied convergence guarantees and reduced memory requirements for variants of Kelley's method [11, 6]. These variants are computationally disadvantaged compared to OSM unless these variants allow approximation of only part of the objective. Among the earliest work on bounded storage in proximal level bundle methods is a paper by Kiwiel (1995) [12]. This method projects iterates onto successive approximations of the level set of the objective; however, unlike our method, it is sensitive to the choice of parameters (level sets) and oblivious to any simplicial structure: iterates are often not extreme points of the epigraph of the function. Subsequent work on the proximal setup uses trust regions, penalty functions, level sets, and other more complex algorithmic tools; we refer the reader to [18] for a survey on bundle methods. For the dual problem, a paper by Von Hohenbalken (1977) [24] shares some elements of our proof techniques. However, their results only apply to differentiable objectives and do not bound the memory. Another restricted simplicial decomposition method was proposed by Hearn et. al. (1987) [10], which limits the constraint set by user-defined parameters (e.g., $r = 1$ reduces to the Frank-Wolfe algorithm [9]): it can replace an atom with minimal weight in the current convex combination with a prior iterate of the algorithm,

which may be strictly inside the feasible region. In contrast, L-FCFW obeys a known upper bound $(n + 1)$ on the number of vertices, and hence requires no parameter tuning.

**Applications:** Composite convex and submodular objectives have gained popularity over the last few years in a large number of machine learning applications such as structured regularization or empirical risk minimization [4]: $\min_{w \in \mathbb{R}^n} \sum_i l(y_i, w^\top x_i) + \lambda \Omega(w)$, where $w$ are the model parameters and $\Omega : \mathbb{R}^n \to \mathbb{R}$ is a regularizer. The Lovász extension can be used to obtain a convex relaxation of a regularizer that penalizes the support of the solution $w$ to achieve structured sparsity, which improves model interpretable or encodes knowledge about the domain. For example, fused regularization uses $\Omega(w) = \sum_i |w_i - w_{i+1}|$, which is the Lovász extension of the generalized cut function, and group regularization uses $\Omega(w) = \sum_g d_g \|w_g\|_\infty$, which is the Lovász extension of the coverage submodular function. (See Appendix A, Table 1 for details on these and other submodular functions.)

Furthermore, minimizing a composite convex and submodular objective is dual to minimizing a convex objective over a submodular polytope (under mild conditions). This duality is central to the present work. First-order projection-based methods like online stochastic mirror descent and its variants require computing a Bregman projection $\min\{\omega(x) + \nabla\omega(y)^\top (x - y) : x \in P\}$ to minimize a strictly convex function $\omega : \mathbb{R}^n \to \mathbb{R}$ over the set $P \subseteq \mathbb{R}^n$. Computing this projection is often difficult, and prevents practical application of these methods, though this class of algorithms is known to obtain near optimal convergence guarantees in various settings [20, 1]. Using L-FCFW to compute these projections can reduce the memory requirements in variants of online mirror descent used for learning over spanning trees to reduce communication delays in networks, [13]), permutations to model scheduling delays [26], and k-sets for principal component analysis [25], to give a few examples of submodular online learning problems. Other example applications of convex minimization over submodular polytopes include computation of densest subgraphs [19], computation of a lower bound for the partition function of log-submodular distributions [7] and distributed routing [14].

**Summary of contributions:** We discuss background and the problem formulations in Section 2. Section 3 describes L-KM, our proposed limited memory version of OSM, and shows that L-KM converges and solves a problem over $\mathbb{R}^n$ using subproblems with at most $n + 1$ constraints. We introduce duality between our primal and dual problems in Section 4. Section 5 introduces a limited memory (and hence faster) version of Fully-Corrective Frank-Wolfe, L-FCFW, and proves linear convergence of L-FCFW. We establish the duality between L-KM and L-FCFW in Appendix E and thereby show L-KM achieves linear convergence and L-FCFW solves subproblems over no more than $n + 1$ vertices. We present preliminary experiments in Section 7 that highlight the reduced memory usage of both L-KM and L-FCFW and show that their performance compares favorably with OSM and FCFW respectively.

## 2 Background and Notation

Consider a ground set $V$ of $n$ elements on which the submodular function $F : 2^V \to \mathbb{R}$ is defined. The function $F$ is said to be submodular if $F(A) + F(B) \geq F(A \cup B) + F(A \cap B)$ for all $A, B \subseteq V$. This is equivalent to the diminishing marginal returns characterization mentioned before. Without loss of generality, we assume $F(\emptyset) = 0$. For $x \in \mathbb{R}^n$, $A \subseteq V$, we define $x(A) = \sum_{k \in A} x(k) = \mathbf{1}_A^\top x$, where $\mathbf{1}_A \in \mathbb{R}^n$ is the indicator vector of $A$, and let both $x(k)$ and $x_k$ denote the $k$th element of $x$.

Given a submodular set function $F : V \to \mathbb{R}$, the submodular polyhedron and the base polytope are defined as $P(F) = \{w \in \mathbb{R}^n : w(A) \leq F(A), \forall A \subseteq V\}$, and $B(F) = \{w \in \mathbb{R}^n : w(V) = F(V), w \in P(F)\}$, respectively. We use $\text{vert}(B(F))$ to denote the vertex set of $B(F)$. The Lovász extension of $F$ is the piecewise linear function [17]

$$f(x) = \max_{w \in B(F)} w^\top x. \tag{1}$$

The Lovász extension can be computed using Edmonds' greedy algorithm for maximizing linear functions over the base polytope (in $O(n \log n + n\gamma)$ time, where $\gamma$ is the time required to compute the submodular function value). This extension can be defined for any set function, however it is convex if and only if the set function is submodular [17]. We call a permutation $\pi$ over $[n]$ *consistent*[1]

with $x \in \mathbb{R}^n$ if $x_{\pi_i} \geq x_{\pi_j}$ whenever $i \leq j$. Each permutation $\pi$ corresponds to an extreme point $x_{\pi_k} = F(\{\pi_1, \pi_2, \ldots, \pi_k\}) - F(\{\pi_1, \pi_2, \ldots, \pi_{k-1}\})$ of the base polytope. For $x \in \mathbb{R}^n$, let $\mathcal{V}(x)$ be the set of vertices $B(F)$ that correspond to permutations consistent with $x$. Note that

$$\partial f(x) = \text{conv}(\mathcal{V}(x)) = \underset{w \,\in\, B(F)}{\text{argmax}}\ w^\top x, \tag{2}$$

where $\partial f(x)$ is the subdifferential of $f$ at $x$ and $\text{conv}(S)$ represents the convex hull of the set $S$.

We assume all convex functions in this paper are closed and proper [21]. Given a convex function $g : \mathbb{R}^n \to \mathbb{R}$, its Fenchel conjugate $g^* : \mathbb{R}^n \to \mathbb{R}$ is defined as

$$g^*(w) \stackrel{\Delta}{=} \max_{x \,\in\, \mathbb{R}^n} w^\top x - g(x). \tag{3}$$

Note that when $g$ is strongly convex, the right hand side of (3) always has an unique solution, so $g^*$ is defined for all $w \in \mathbb{R}^n$. Fenchel conjugates are always convex, regardless of the convexity of the original function. Since we assume $g$ is closed, $g^{**} = g$. Fenchel conjugates satisfy $(\partial g)^{-1} = \partial g^*$ in the following sense:

$$w \in \partial g(x) \iff g(x) + g^*(w) = w^\top x \iff x \in \partial g^*(w), \tag{4}$$

where $\partial g(x)$ is the subdifferential of $g$ at $x$. When $g$ is $\alpha$-strongly convex and $\beta$-smooth, $g^*$ is $1/\beta$-strongly convex and $1/\alpha$-smooth [21, Section 4.2]. (See Appendix A.2 for details.)

Proofs of all results that do not follow easily from the main text can be found in the appendix.

## 3 Limited Memory Kelley's Method

We now present our novel limited memory adaptation L-KM of the Original Simplicial Method (OSM). We first briefly review OSM as proposed by Bach [3, Section 7.7] and discuss problems of OSM with respect to memory requirements and the rate of convergence. We then highlight the changes in OSM, and verify that these changes will enable us to show a bound on the memory requirements while maintaining finite convergence. Proofs omitted from this section can be found in Appendix C.

**Original Simplicial Method:** To solve the primal problem (P), it is natural to approximate the piecewise linear Lovász extension $f$ with cutting planes derived from the function values and subgradients of the function at previous iterations, which results in piecewise linear lower approximations of $f$. This is the basic idea of OSM introduced by Bach in [3]. This approach contrasts with Kelley's Method, which approximates the entire objective function $g + f$. OSM adds a cutting plane to the approximation of $f$ at each iteration, so the number of the linear constraints in its subproblems grows linearly with the number of iterations.[2] Hence it becomes increasingly challenging to solve the subproblem as the number of iterations grows up. Further, in spite of a finite convergence, as mentioned in the introduction there was no known rate of convergence for OSM or its dual method prior to this work.

**Limited Memory Kelley's Method:** To address these two challenges — memory requirements and unknown convergence rate — we introduce and analyze a novel limited memory version L-KM of OSM which ensures that the number of cutting planes maintained by the algorithm at any iteration is bounded by $n + 1$. This thrift bounds the size of the subproblems at any iteration, thereby making L-KM cheaper to implement. We describe L-KM in detail in Algorithm 1.

L-KM and OSM differ only in the set of vertices $\mathcal{V}^{(i)}$ considered at each step: L-KM keeps only those vectors $w \in \mathcal{V}^{(i-1)}$ that maximize $w^\top x^{(i)}$, whereas OSM keeps every vector in $\mathcal{V}^{(i-1)}$.

We state some properties of L-KM here with proofs in Appendix C. We will revisit many of these properties later via the lens of duality.

The sets $\mathcal{V}^{(i)}$ in L-KM are affinely independent, which shows the size of the subproblems is bounded.

**Theorem 1.** *For all $i \geq 0$, vectors in $\mathcal{V}^{(i)}$ are affinely independent. Moreover, $|\mathcal{V}^{(i)}| \leq n + 1$.*

**Algorithm 1** L-KM: The Limited Memory Kelley's Method for ($P$)

---

**Require:** strongly convex function $g : \mathbb{R}^n \to \mathbb{R}$, submodular function $F : 2^n \to \mathbb{R}$, tolerance $\epsilon \geq 0$
**Ensure:** $\epsilon$-suboptimal solution $x^\sharp$ to ($P$)
1: initialize: choose $x^{(0)} \in \mathbb{R}^n$, set $\emptyset \subset \mathcal{V}^{(0)} \subseteq \text{vert}(B(F))$ affinely independent
2: **for** $i = 1, 2, \dots$ **do**
3:    **Convex subproblem.** Define approximation $f_{(i)}(x) = \max\{w^\top x : w \in \mathcal{V}^{(i-1)}\}$ and solve

$$x^{(i)} = \arg\min g(x) + f_{(i)}(x).$$

4:    **Submodular subproblem**. Compute value and subgradient of $f$ at $x^{(i)}$

$$f(x^{(i)}) = \max_{w \in B(F)} w^\top x^{(i)}, \qquad v^{(i)} \in \partial f(x^{(i)}) = \underset{w \in B(F)}{\arg\max}\, w^\top x^{(i)}.$$

5:    **Stopping condition.** Break if duality gap $p^{(i)} - d^{(i)} \leq \epsilon$, where

$$p^{(i)} = g(x^{(i)}) + f(x^{(i)}), \qquad d^{(i)} = g(x^{(i)}) + f_{(i)}(x^{(i)}).$$

6:    **Update memory.** Identify active set $\mathcal{A}^{(i)}$ and update memory $\mathcal{V}^{(i)}$:

$$\mathcal{A}^{(i)} = \{w \in \mathcal{V}^{(i-1)} : w^\top x^{(i)} = f_{(i)}(x^{(i)})\}, \qquad \mathcal{V}^{(i)} = \mathcal{A}^{(i)} \cup \{v^{(i)}\}.$$

7: **return** $x^{(i)}$

---

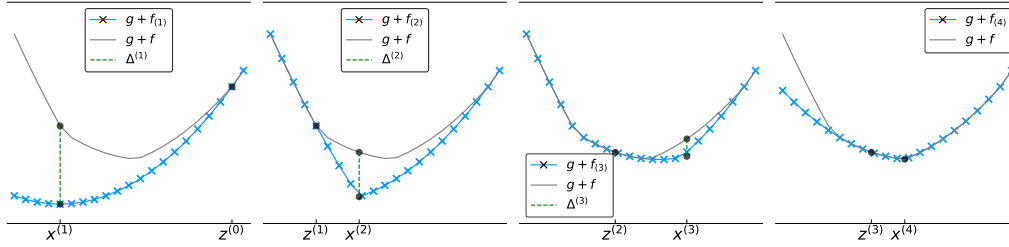

Figure 1: An illustration of L-KM (a)-(d) (left to right): blue curve marked with $\times$ denotes the $i$th function approximation $g + f_{(i)}$. In (d), note that L-KM approximation $g + f_{(4)}$ is obtained by dropping the leftmost constraint in $g + f_{(3)}$ (in (c)), unlike OSM.

L-KM may discard pieces of the lower approximation $f_{(i)}$ at each iteration. However, it does so without any adverse affect on the solution to the current subproblem:

**Lemma 1.** *The convex subproblem (in Step 3 of algorithm* L-KM*) has the same solution and optimal value over the new active set $\mathcal{A}^{(i)}$ as over the memory $\mathcal{V}^{(i-1)}$:*

$$x^{(i)} = \arg\min g(x) + \max\{w^\top x : w \in \mathcal{V}^{(i-1)}\} = \arg\min g(x) + \max\{w^\top x : w \in \mathcal{A}^{(i)}\}.$$

Lemma 1 shows that L-KM remembers the important information about the solution, i.e. only the tight subgradients, at each iteration. Note that at the $i$th iteration, the solution $x^{(i)}$ is unique by the strong convexity of $g$, and thus we can improve the lower bound $d^{(i)}$ since new information (i.e. $v^{(i)}$) is added:

**Corollary 1.** *The sequence $\{d^{(i)}\}$ constructed by* L-KM *form strictly increasing lower bounds on the value of* ($P$): $d^{(1)} < d^{(2)} < \cdots \leq p^\star \overset{\Delta}{=} \min_{x \in \mathbb{R}^n} f(x) + g(x)$.*

**Remark.** *It is easy to see that the sequence $\{p^{(i)}\}$ constructed by* L-KM *form upper bounds of $p^\star$, hence by Corollary 1, $\{p^{(i)} - d^{(i)}\}$ form valid optimality gaps for* L-KM.

**Corollary 2.** L-KM *does not stall: for any iterations $i_1 \neq i_2$, we solve subproblems over a distinct set of vertices $\mathcal{V}^{(i_1)} \neq \mathcal{V}^{(i_2)}$.*

We can strengthen Corollary 2 and show L-KM in fact converges to the exact solution in finite iterations:

**Theorem 2.** L-KM *(Algorithm 1) terminates after finitely many iterations. Moreover, for any given $\epsilon \geq 0$, suppose* L-KM *terminates when $i = i_\epsilon$, then $p^\star + \epsilon \geq p^{(i_\epsilon)} \geq p^\star$ and $p^\star \geq d^{(i_\epsilon)} \geq p^\star - \epsilon$.*

In particular, when we choose $\epsilon = 0$, we have $p^{(i_0)} = p^\star = d^{(i_\epsilon)}$, and $x^{(i_\epsilon)}$ is the unique optimal solution to ($P$).

In this section, we have shown that L-KM solves a series of limited memory convex subproblems with no more than $n + 1$ linear constraints, and produces strictly increasing lower bounds that converge to the optimal value.

## 4 Duality

L-KM solves a series of subproblems parametrized by the sets $\mathcal{V} \subseteq \text{vert}(B(F))$:

$$
\begin{array}{ll}
\text{minimize} & g(x) + t \\
\text{subject to} & t \geq v^\top x, \quad v \in \mathcal{V}
\end{array}
\tag{$P_\mathcal{V}$}
$$

Notice that when $\mathcal{V} = \text{vert}(B(F))$, we recover ($P$). We now analyze these subproblems via duality. The Lagrangian of this problem with dual variables $\lambda_v$ for $v \in \mathcal{V}$ is,

$$
\mathcal{L}(x, t, \lambda) = g(x) + t + \sum_{v \in \mathcal{V}} \lambda_v (v^\top x - t).
$$

The pair $((x, t), \lambda)$ are primal-dual optimal for this problem *iff* they satisfy the KKT conditions [5]:

- *Optimality.*

$$
0 \in \partial_x \mathcal{L}(x, t, \lambda) \implies \sum_{v \in \mathcal{V}} \lambda_v v \in -\partial g(x), \qquad 0 = \frac{d}{dt} \mathcal{L}(x, t, \lambda) \implies \sum_{v \in \mathcal{V}} \lambda_v = 1.
$$

- *Primal feasibility.* $t \geq v^\top x$ for each $v \in \mathcal{V}$.
- *Dual feasibility.* $\lambda_v \geq 0$ for each $v \in \mathcal{V}$.
- *Complementary slackness.* $\lambda_v (v^\top x - t) = 0$ for each $v \in \mathcal{V}$.

The requirement that $\lambda$ lie in the simplex emerges naturally from the optimality conditions, and reduces the Lagrangian to $\mathcal{L}(x, \lambda) = g(x) + \left( \sum_{v \in \mathcal{V}} \lambda_v v \right)^\top x$. One can introduce the variable $w = \sum_{v \in \mathcal{V}} \lambda_v v \in \text{conv}(\mathcal{V})$, which is dual feasible so long as $w \in \text{conv}(\mathcal{V})$. We can rewrite the Lagrangian in terms of $x$ and $w \in \text{conv}(\mathcal{V})$ as $L(x, w) = g(x) + w^\top x$. Minimizing $L(x, w)$ over $x$, we obtain the dual problem

$$
\begin{array}{ll}
\text{maximize} & -g^*(-w) \\
\text{subject to} & w \in \text{conv}(\mathcal{V}).
\end{array}
\tag{$D_\mathcal{V}$}
$$

Note ($D$) is the same as ($D_\mathcal{V}$) if $\mathcal{V} = \text{vert}(B(F))$ and $h(w) = -g^*(-w)$, the Fenchel conjugate of $g$. Notice that $g^*$ is smooth if $g$ is strongly convex (Lemma 4 in Appendix A.2).

**Theorem 3** (Strong Duality). *The primal problem ($P_\mathcal{V}$) and the dual problem ($D_\mathcal{V}$) have the same finite optimal value.*

By analyzing the KKT conditions, we obtain the following result, which we will used later in the design of our algorithms.

**Lemma 2.** *Suppose $(x, \lambda)$ solve (($P_\mathcal{V}$), ($D_\mathcal{V}$)) and $t = \max_{v \in \mathcal{V}} v^\top x$. By complementary slackness,*

$$
\lambda_v > 0 \implies v^\top x = t,
$$
$$
\text{and in particular,} \quad \{v : \lambda_v > 0\} \quad \subseteq \quad \{v : v^\top x = t\}.
$$

Notice $\{v : v^\top x = t\}$ is the active set of L-KM. We will see $\{v : \lambda_v > 0\}$ is the (minimal) active set of the dual method L-FCFW. (If strict complementary slackness holds, these sets are the same.)

The first KKT condition shows how to move between primal and dual optimal variables.

**Theorem 4.** *If $g : \mathbb{R}^n \to \mathbb{R}$ is strongly convex and $w^\star$ solves ($D_\mathcal{V}$), then*

$$
x^\star = (\partial g)^{-1}(-w^\star) = \nabla g^*(-w^\star)
\tag{5}
$$

*solves ($P_\mathcal{V}$). If in addition $g$ is smooth and $x^\star$ solves ($P_\mathcal{V}$), then $w^\star = \nabla g(x^\star)$ solves ($D_\mathcal{V}$).*

---

**Algorithm 2** L-FCFW: Limited Memory Fully Corrective Frank Wolfe for ($D$)

---

**Require:** smooth concave function $h : \mathbb{R}^n \to \mathbb{R}$, submodular function $F : 2^n \to \mathbb{R}$, tolerance $\epsilon \geq 0$
**Ensure:** $\epsilon$-suboptimal solution $w^\sharp$ to ($D$)
 1: initialize: set $\emptyset \subset \mathcal{V}^{(0)} \subseteq \text{vert}(B(F))$
 2: **for** $i = 1, 2, \ldots$ **do**
 3:     **Convex subproblem.** Solve

$$w^{(i)} = \text{argmax}\{h(w) : w \in \text{conv}(\mathcal{V}^{(i-1)})\}.$$

   For each $v \in \mathcal{V}^{(i)}$, define $\lambda_v \geq 0$ so that $w^{(i)} = \sum_{v \in \mathcal{V}^{(i)}} \lambda_v v$ and $\sum_{v \in \mathcal{V}^{(i)}} \lambda_v = 1$.
 4:     **Submodular subproblem.** Compute gradient $x^{(i)} = \nabla h(w^{(i)})$ and solve

$$v^{(i)} = \text{argmax}\{w^\top x^{(i)} : w \in B(F)\}.$$

 5:     **Stopping condition.** Break if duality gap $p^{(i)} - d^{(i)} \leq \epsilon$, where

$$p^{(i)} = (v^{(i)})^\top x^{(i)}, \qquad d^{(i)} = (w^{(i)})^\top x^{(i)}.$$

 6:     **Update memory.** Identify a supserset of active vertices $\mathcal{B}^{(i)}$ and update memory $\mathcal{V}^{(i)}$:

$$\mathcal{B}^{(i)} \supseteq \{w \in \mathcal{V}^{(i-1)} : \lambda_w > 0\} \qquad \mathcal{V}^{(i)} = \mathcal{B}^{(i)} \cup \{v^{(i)}\}.$$

 7: **return** $w^{(i)}$

---

*Proof.* Check the optimality conditions to prove the result. By definition, $x^\star$ satisfies the first optimality condition. To check complementary slackness, we rewrite the condition as

$$\lambda_v(v^\top x - t) = 0 \quad \forall v \in \mathcal{V} \iff \left(\sum_{v \in \mathcal{V}} \lambda_v v\right)^\top x = t \iff w^\top x = \max_{v \in \mathcal{V}} v^\top x.$$

Notice $(w^\star)^\top (\nabla g^*(-w^\star)) = \max_{v \in \mathcal{V}} v^\top (\nabla g^*(-w^\star))$ by optimality of $w^\star$, since $v - w^\star$ is a feasible direction for any $v \in \mathcal{V}$, proving $x^\star = \nabla g^*(-w^\star)$ solves ($P_\mathcal{V}$).

That the primal optimal variable yields a dual optimal variable via $w^\star = \nabla g(x^\star)$ follows from a similar argument together with ideas from the proof of strong duality in Appendix D. $\qquad \square$

## 5   Solving the dual problem

Let's return to the dual problem ($D$): maximize a smooth concave function $h(w) = -g^*(-w)$ over the polytope $B(F) \subseteq \mathbb{R}^n$. Linear optimization over this polytope is easy; hence a natural strategy is to use the Frank-Wolfe method or one of its variants [15]. However, since the cost of each linear minimization is not negligible, we will adopt a Frank-Wolfe variant that makes considerable progress at each iteration by solving a subproblem of moderate complexity: LIMITED MEMORY FULLY CORRECTIVE FRANK-WOLFE (L-FCFW, Algorithm 2), which at every iteration exactly minimizes the function $-g^*(-w)$ over the the convex hull of the current subset of vertices $\mathcal{V}^{(i)}$. Here we overload notation intentionally: when $g$ is smooth and strongly convex, we will see that we can choose the set of vertices $\mathcal{V}^{(i)}$ in L-FCFW (Algorithm 2) so that the algorithm matches either L-KM or OSM depending on the choice of $\mathcal{B}^{(i)}$ (Line 6 of L-FCFW). For details of the duality between L-KM and L-FCFW see Section 6.

**Limited memory.**   In L-FCFW, we may choose any active set $\mathcal{B}^{(i)} \supseteq \{w \in \mathcal{V}^{(i-1)} : \lambda_w > 0\}$. When $\mathcal{B}^{(i)} = \mathcal{V}^{(i-1)}$, we call the algorithm (vanilla) FCFW. When $\mathcal{B}^{(i)}$ is chosen to be small, we call the algorithm LIMITED MEMORY FCFW (L-FCFW). Standard FCFW increases the size of the active set at each iteration, whereas the most limited memory variant of L-FCFW uses only those vertices needed to represent the iterate $w^{(i)}$.

Moreover, recall Carathéodory's theorem (see e.g. [23]): for any set of vectors $\mathcal{V}$, if $x \in \text{conv}(\mathcal{V}) \subseteq \mathbb{R}^n$, then there exists a subset $A \subseteq \mathcal{V}$ with $|A| \leq n+1$ such that $x \in \text{conv}(A)$. Hence we see we can choose $\mathcal{B}^{(i)}$ to contain at most $n+1$ vertices at each iteration (hence $n+2$ in $\mathcal{V}^{(i)}$), or even fewer if the iterate lies on a low-dimensional face of $B(F)$. (The size of $\mathcal{B}^{(i)}$ may depend on the solver used

for ($D_\mathcal{V}$); to reduce the size of $\mathcal{B}^{(i)}$, we can minimize a random linear objective over the optimal set of ($D_\mathcal{V}$) as in [22].)

**Linear convergence.** Lacoste-Julien and Jaggi [15] show that FCFW converges linearly to an $\epsilon$-suboptimal solution when $g$ is smooth and strongly convex so long as the active set $\mathcal{B}^{(i)}$ and iterate $x^{(i)}$ satisfy three conditions they call *approximate correction*($\epsilon$):

1. **Better than** FW. $h(y^{(i)}) \leq \min_{\lambda \in [0,1]} h((1-\lambda)w^{(i-1)} + \lambda v^{(i-1)}))$.

2. **Small away-step gap.** $\max\{(w^{(i)} - v)^\top x^{(i)} : v \in \mathcal{V}(w^{(i)})\} \leq \epsilon$, where $\mathcal{V}(w^{(i)}) = \{v \in \mathcal{V}^{(i-1)} : \lambda_v > 0\}$.

3. **Representation.** $x^{(i)} \in \mathrm{conv}(\mathcal{B}^{(i)})$.

By construction, iterates of L-FCFW always satisfy these conditions with $\epsilon = 0$:

1. **Better than FW.** For any $\lambda \in [0,1]$, $w = (1-\lambda)w^{(i-1)} + \lambda v^{(i-1)}$ is feasible.

2. **Zero away-step gap.** For each $v \in \mathcal{V}^{(i)}$, if $w^{(i)} = v$, then clearly $(w^{(i)} - v)^\top(x^{(i)}) = 0$. otherwise (if $w^{(i)} \neq v$) $v - w^{(i)}$ is a feasible direction, and so by optimality of $w^{(i)}$ $(w^{(i)} - v)^\top(x^{(i)}) \leq 0$.

3. **Representation.** We have $w^{(i)} \in \mathrm{conv}(\mathcal{B}^{(i)})$ by construction of $\mathcal{B}^{(i)}$.

Hence we have proved Theorem 5:

**Theorem 5.** *Suppose $g$ is $\alpha$-smooth and $\beta$-strongly convex. Let $M$ be the diameter of $B(F)$ and $\delta$ be the pyramidal width[3] of $P$, then the lower bounds $d^{(i)}$ in* L-FCFW *(Algorithm 2) converges linearly at the rate of $1 - \rho$, i.e. $p^\star - d^{(i+1)} \leq (1-\rho)(p^\star - d^{(i)})$, where $\rho \triangleq \frac{\beta}{4\alpha}(\frac{\delta}{M})^2$.*

**Primal-from-dual algorithm.** Recall that dual iterates yield primal iterates via Theorem 4. Hence the gradients $x^{(i)} = -\nabla g^*(-w^{(i)})$ computed by L-FCFW converge linearly to the solution $x^\star$ of ($P$). However, it is difficult to run L-FCFW directly to solve ($D$) given only access to $g$, since in that case computing $g^*$ and its gradient requires solving another optimization problem; moreover, we will see below that L-KM computes the same iterates. See Appendix E for more discussion.

# 6 L-KM (and OSM) converge linearly

L-KM (Algorithm 1) and L-FCFW (Algorithm 2) are dual algorithms in the following strong sense:

**Theorem 6.** *Suppose $g$ is $\alpha$-smooth and $\beta$-strongly convex. In* L-FCFW *(Algorithm 2), suppose we choose $\mathcal{B}^{(i)} = \mathcal{A}^{(i)} = \{v \in \mathcal{V}^{(i-1)} : v^\top x^{(i)} = {w^{(i)}}^\top x^{(i)}\}$. Then*

1. *The primal iterates $x^{(i)}$ of* L-KM *and* L-FCFW *match.*

2. *The sets $\mathcal{V}^{(i)}$ used at each iteration of* L-KM *and* L-FCFW *match.*

3. *The upper and lower bounds $p^{(i)}$ and $d^{(i)}$ of* L-KM *and* L-FCFW *match.*

**Corollary 3.** *The active sets of* L-FCFW *can be chosen to satisfy $|\mathcal{B}^{(i)}| \leq n + 1$.*

**Theorem 7.** *Suppose $g$ is $\alpha$-strongly convex and let $M$ be the diameter of $B(F)$, the duality gap $p^{(i)} - d^{(i)}$ in* L-KM *(Algorithm 1) converges linearly: $p^{(i)} - d^{(i)} \leq (p^\star - d^{(i)}) + M^2/(2\beta)$ when $(p^\star - d^{(i)}) \geq M^2/(2\beta)$ and $p^{(i)} - d^{(i)} \leq M\sqrt{2(p^\star - d^{(i)})/\beta}$ otherwise. Note that $p^\star - d^{(i)}$ converges linearly by Theorem 5.*

When $g$ is smooth and strongly convex, OSM and vanilla FCFW are dual algorithms in the same sense when we choose $\mathcal{B}^{(i)} = \mathcal{V}^{(i-1)}$. For details of the duality between OSM and L-FCFW see Appendix G. Hence we have a similar convergence result for OSM:

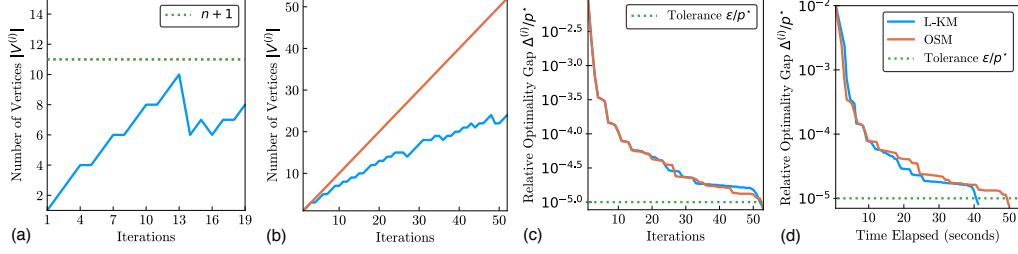

Figure 2: Dimension $n = 10$ in (a), $n = 100$ in (b), (c) and (d). The methods converged in (a), (b), (c) and (d).

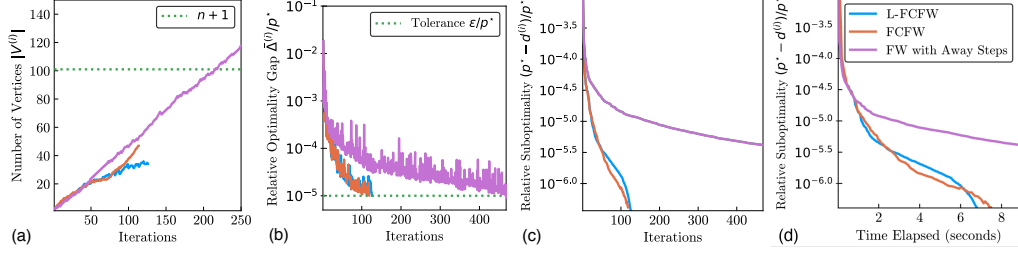

Figure 3: L-FCFW and FCFW converged in all plots, FW with away steps has converged in (b), (c) and (d).

**Theorem 8.** *Suppose $g$ is $\alpha$-strongly convex and let $M$ be the diameter of $B(F)$, the duality gap $p^{(i)} - d^{(i)}$ in OSM converges linearly: $p^{(i)} - d^{(i)} \leq (p^{\star} - d^{(i)}) + M^2/(2\beta)$ when $(p^{\star} - d^{(i)}) \geq M^2/(2\beta)$ and $p^{(i)} - d^{(i)} \leq M\sqrt{2(p^{\star} - d^{(i)})/\beta}$ otherwise.*

**Remark.** *Note that $p^{\star} - d^{(i)}$ converges linearly by Theorem 5, Theorem 7 and Theorem 7 imply* L-KM *and* OSM *converge linearly when $g$ is smooth and strongly convex.*

Moreover, this connection generates a new way to prune the active set of L-KM even further using a primal dual solver: we may use any active set $\mathcal{B}^{(i)} \supseteq \{w \in \mathcal{V}^{(i-1)} : \lambda_w > 0\}$, where $\lambda \in \mathbb{R}^{|\mathcal{V}^{(i-1)}|}$ is a dual optimal solution to ($P_\mathcal{V}$). When strict complementary slackness fails, we can have $\mathcal{B}^{(i)} \subset \mathcal{A}^{(i)}$.

## 7 Experiments and Conclusion

We present in this section a computational study: we minimize non-separable composite functions $g + f$ where $g(x) = x^\top (A + n\mathbf{I}_n)x + b^\top x$ for $x \in \mathbb{R}^n$, and $f$ is the Lovász extension of the submodular function $F(A) = \frac{|A|(2n-|A|+1)}{2}$ for $A \subseteq [n]$. To construct $g(\cdot)$, entries of $A \in M_n$ and $b \in \mathbb{R}^n$ were randomly sampled from $U[-1,1]^{n \times n}$, and $U[0,n]^n$ respectively. $\mathbf{I}_n$ is an $n \times n$ identity matrix. We remark that L-KM converges so quickly that the bound on the size of the active set is less important, in practice, than the fact that the active set need not grow at every iteration.

**Primal convergence**: We first solve a toy problem for $n = 10$ and show that the number of constraints does not exceed $n + 1$. Note that the number of constraints might oscillate before it reaches $n + 1$ (Fig. 2(a)). We next compare the memory used in each iteration (Fig. 2(b)), the optimality gap per iteration (Fig. 2(c)), and the running time (Fig. 2(d)) of L-KM and OSM by solving the problem for $n = 100$ up to accuracy of $10^{-5}$ of the optimal value. Note that L-KM uses much less memory compared to OSM, converges at almost the same rate in iterations, and its running time per iteration improves as the iteration count increases.

**Dual convergence**: We compare the convergence of L-FCFW, FCFW and Frank-Wolfe with away steps for the dual problem $\max_{w \in B(F)} -(-w - b)^\top (A + n\mathbf{I}_n)^{-1}(-w - b)$ for $n = 100$ up to relative accuracy of $10^{-5}$. L-FCFW maintains smaller sized subproblems (Figure (3)(a)), and it converges faster than FCFW as the number of iteration increases (Figure (3)(d)). Their provable duality gap converges linearly in the number of iterations. Moreover, as shown in Figures (3)(b) and (c), L-FCFW and FCFW return better approximate solutions than Frank-Wolfe with away steps under the same optimality gap tolerance.

**Conclusion** This paper defines a new limited memory version of Kelley's method adapted to composite convex and submodular objectives, and establishes the first convergence rate for such a method,

solving the open problem proposed in [2, 3]. We show bounds on the memory requirements and convergence rate, and demonstrate compelling performance in practice.

**Acknowledgments**

This work was supported in part by DARPA Award FA8750-17-2-0101. A part of this work was done while the first author was at the Department of Mathematical Sciences, Tsinghua University and while the second author was visiting the Simons Institute, UC Berkeley. The authors would also like to thank Sebastian Pokutta for invaluable discussions on the Frank-Wolfe algorithm and its variants.

## Footnotes

[1]Therefore, the Lovász extension can also be written as $f(x) = \sum_k x_{\pi_k}[F(\{\pi_1, \pi_2, \ldots, \pi_k\}) - F(\{\pi_1, \pi_2, \ldots, \pi_{k-1}\})]$ where $\pi$ is a permutation consistent with $x$ and $F(\emptyset) = 0$ by assumption.

[2] Concretely, we obtain OSM from Algorithm 1 by setting $\mathcal{V}^{(i)} \stackrel{\Delta}{=} \mathcal{V}^{(i-1)} \cup \{v^{(i)}\}$ in step 6.

[3]See Appendix H for definitions of the diameter and pyramidal width.

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
