[Supplementary Material]

# A Additional Background

## A.1 Additional Examples

We list some examples of popular submodular functions in Table 1.

| Problem | Submodular function, $S \subseteq E$ (unless specified) |
|---|---|
| n experts (simplex), $E = \{1, \ldots, n\}$ | $f(S) = 1$ |
| k out of n experts (k-simplex), $E = \{1, \ldots, n\}$ | $f(S) = \min\{|S|, k\}$ |
| Permutations over $E = \{1, \ldots, n\}$ | $f(S) = \sum_{s=1}^{|S|}(n+1-s)$ |
| k-truncated permutations over $E = \{1, \ldots, n\}$ | $f(S) = (n-k)|S|$ for $|S| \leq k$, $f(S) = k(n-k) + \sum_{s=k+1}^{|S|}(n+1-s)$ if $|S| \geq k$ |
| Spanning trees on $G = (V, E)$ | $f(S) = |V(S)| - \kappa(S)$, $\kappa(S)$ is the number of connected components of $S$ |
| Matroids over ground set $E$: $M = (E, (\mathcal{I})), (\mathcal{I}) \subseteq 2^E$ | $f(S) = r_M(S)$, the rank function of the matroid |
| Coverage of T: given $T_1, \ldots, T_n \subseteq T$ | $f(S) = |\bigcup_{i \in S} T_i|$, $E = \{1, \ldots, n\}$ |
| Cut functions on a directed graph $D = (V, E)$, $c : E \to \mathbb{R}_+$ | $f(S) = c(\delta^{out}(S))$, $S \subseteq V$ |
| Flows into a sink vertex $t$, given a directed graph $D = (V, E)$ and costs $c : E \to \mathbb{R}_+$ | $f(S) = $ max flow from $S \subseteq V \setminus \{t\}$ into $t$ |
| Maximal elements in $E$, $h : E \to \mathbb{R}$ | $f(S) = \max_{e \in S} h(e)$, $f(\emptyset) = \min_{e \in E} h(e)$ |
| Entropy $H$ of random variables $X_1, \ldots, X_n$ | $f(S) = H(\bigcup_{i \in S} X_i)$, $E = \{1, \ldots, n\}$ |

Table 1: Problems and the submodular functions (on ground set of elements $E$) that give rise to them.

## A.2 Strong Convexity and Smoothness

We say a function $g : \mathbb{R}^n \to \mathbb{R}$ is $\alpha$-strongly convex if $g(x) - \alpha/2\|x\|^2$ is convex, where $\alpha > 0$. It is easy to see that the sum of a stronly convex function and a piecewise linear function is still strongly convex, and we have

**Lemma 3.** *When $g : \mathbb{R}^n \to \mathbb{R}$ is a strongly convex function, then*

$$\text{minimize} \quad g(x) + \max_{w \, \in \, \text{conv}(\mathcal{V})} w^\top x \tag{$P_\mathcal{V}$}$$

*has a unique optimal solution $x^\star$ for all $\mathcal{V} \subseteq \mathbb{R}^n$.*

On the other hand, we say a function $g : \mathbb{R}^n \to \mathbb{R}$ is $\beta$-smooth if there exists $\beta > 0$ such that $g(x) - \beta/2\|x\|^2$ is concave. We have[21]:

**Lemma 4.** *When a function $g$ is $\alpha$-strongly convex, its Fenchel conjugate $g^*$ is $\frac{1}{\alpha}$- smooth.*

**Lemma 5.** *When a function $g$ is $\beta$-smooth, its Fenchel conjugate $g^*$ is $\frac{1}{\beta}$-strongly convex.*

# B The Original Simplicial Method (Section 3)

We present the Original Simplicial Method (OSM) in Algorithm 3.

# C Limited Memory Kelley's Method (Section 3)

In this section, we provide proofs of some of the results in Section 3.

**Proof of Theorem 1.**

*Proof.* We prove this by induction. The claim is true for $i = 0$ since $\mathcal{V}^{(0)}$ has only one element. Suppose that the claim is true for $i < i_0$. When $\Delta > \epsilon$, we have $v^{(i_0)\top} x^{(i_0)} = f(x^{(i_0)}) > f_{(i_0)}(x^{(i_0)})$. From $\mathcal{A}^{(i_0)} \subseteq \{w \in \mathbb{R}^n \mid w^\top x^{(i_0)} = f_{(i_0)}(x^{(i_0)})\}$ we have $v^{(i_0)} \notin \text{affine}(\mathcal{A}^{(i_0)})$. Otherwise when $\Delta^{(i)} \leq \epsilon$, the algorithm terminates in the $i_0$th iteration.

Since vectors in $\mathcal{V}^{(i)}$ are affinely independent, we have $|\mathcal{V}^{(i)}| \leq n + 1$ for all $i$ since $\mathcal{V}^{(i)} \subseteq \mathbb{R}^n$. $\qquad\square$

Before proving Lemma 1, we first present a lemma that is used in the proof of Lemma 1:

---
**Algorithm 3** OSM: The Original Simplicial Method for ($P$)

---
**Require:** strongly convex function $g : \mathbb{R}^n \to \mathbb{R}$, submodular function $F : 2^n \to \mathbb{R}$, tolerance $\epsilon > 0$
**Ensure:** $\epsilon$-suboptimal solution $x^\sharp$
 1: initialize: choose $x^{(0)} \in \mathbb{R}^n$, set $\mathcal{V}^{(0)} = \emptyset$
 2: **for** $i = 1, 2, \ldots$ **do**
 3:     **Convex subproblem.** Define approximation $f_{(i)}(x) = \max\{w^\top x : w \in \mathcal{V}^{(i-1)}\}$ and solve

$$x^{(i)} = \operatorname{argmin} g(x) + f_{(i)}(x).$$

 4:     **Submodular subproblem.** Compute value and subgradient of $f$ at $x^{(i)}$

$$f(x^{(i)}) = \max_{w \in B(F)} w^\top x^{(i)}, \qquad v^{(i)} \in \partial f(x^{(i)}) = \operatorname*{argmax}_{w \in B(F)} w^\top x^{(i)}.$$

 5:     **Stopping condition.** Break if duality gap $p^{(i)} - d^{(i)} \leq \epsilon$, where

$$p^{(i)} = g(x^{(i)}) + f(x^{(i)}), \qquad d^{(i)} = g(x^{(i)}) + f_{(i)}(x^{(i)}).$$

 6:     **Update memory.** Update memory $\mathcal{V}^{(i)}$:

$$\mathcal{V}^{(i)} = \mathcal{V}^{(i-1)} \cup \{v^{(i)}\}.$$

 7: **return** $x^{(i)}$

---

**Lemma 6.** *Given a submodular function $F : 2^V \to \mathbb{R}$, let $\mathcal{W} \subseteq \operatorname{vert}(B(F))$ be a subset of the vertices of its base polytope. For the piecewise linear function*

$$\tilde{f}(x) = \max_{w \,\in\, \operatorname{conv}(\mathcal{W})} w^\top x,$$

*let $\mathcal{A}(x) \triangleq \{w^\star \in \mathcal{W} \mid {w^\star}^\top x = \tilde{f}(x)\}$ be the points in $\mathcal{W}$ that are active at $x$. Then given any $\bar{x} \in \mathbb{R}^n$, there exists $\epsilon > 0$ such that*

$$\tilde{f}(x) = \max_{w^\star \,\in\, \operatorname{conv}(\mathcal{A}(\bar{x}))} {w^\star}^\top x$$

*for all $x \in \mathcal{B}(\bar{x}, \epsilon)$.*

*Proof.* Since $\mathcal{W}$ is finite, we have $\tilde{f}(\bar{x}) \geq \max_{\tilde{w} \in \mathcal{W} \setminus \mathcal{A}(\bar{x})} \tilde{w}^\top \bar{x} + \epsilon$, where $\epsilon > 0$. Let $L = \max_{w \in \mathcal{W}} \|w\|$, then for all $x \in \mathcal{B}(\bar{x}, \epsilon/(3L))$, $w^\star \in \mathcal{A}(\bar{x})$ and $\tilde{w} \in \mathcal{W} \setminus \mathcal{A}(\bar{x})$, we have

$$
\begin{aligned}
{w^\star}^\top x - \tilde{w}^\top x &= (w^\star - \tilde{w})^\top \bar{x} + {w^\star}^\top (x - \bar{x}) + \tilde{w}^\top (\bar{x} - x) \\
&\geq \epsilon - L\frac{\epsilon}{3L} - L\frac{\epsilon}{3L} \\
&= \frac{\epsilon}{3}.
\end{aligned}
\tag{6}
$$

Hence $\tilde{f}(x) > \tilde{w}^\top x$ for all $x \in \mathcal{B}(\bar{x}, \epsilon/(3L))$ and $\tilde{w} \in \mathcal{W} \setminus \mathcal{A}(\bar{x})$, which is equivalent to $\tilde{f}(x) = \max_{w^\star \in \operatorname{conv}(\mathcal{A}(\bar{x}))} {w^\star}^\top x$ for all $x \in \mathcal{B}(\bar{x}, \epsilon/(3L))$. $\qquad\square$

**Proof of Lemma 1.**

*Proof.* Let $P_{(i)}(x) \triangleq \min g(x) + \max_{w \,\in\, \operatorname{conv}(\mathcal{V}^{(i-1)})} w^\top x = g(x) + f_{(i)}(x)$ and $\widetilde{P}_{(i)} \triangleq \min_{x \,\in\, \mathbb{R}^n} g(x) + \max_{w \,\in\, \operatorname{conv}(\mathcal{A}^{(i)})} w^\top x$. There exists at least one $w^\star \in \mathcal{V}^{(i-1)}$ such that $f_{(i)}(x^{(i)}) = {w^\star}^T x^{(i)}$. Therefore, $P_{(i)}(x^{(i)}) = g(x^{(i)}) + {w^\star}^T x^{(i)} = g(x^{(i)}) + \max_{w \in \operatorname{conv}(\mathcal{A}^{(i)})} w^\top x^{(i)} = \widetilde{P}_{(i)}(x^{(i)})$, where the last equality follows from the definition of $\mathcal{A}^{(i)}$. Next, if we can show local optimality of $x^{(i)}$ for $\widetilde{P}_{(i)}$, this would imply global optimality of $x^{(i)}$ for $\widetilde{P}_{(i)}$ due to convexity of $\widetilde{P}_{(i)}$, thus $P_{(i)}$ and $\widetilde{P}_{(i)}$ will have the same optimal value. By the definition of $\mathcal{A}^{(i)}$ and Lemma 6, we have $f(x) = \max_{w \in \operatorname{conv}(\mathcal{A}(x^{(i)}))} = f_{(i)}(x^{(i)})$ in $\mathcal{B}(x^{(i)}, \epsilon)$ for some $\epsilon > 0$. Thus $P_{(i)}(x^{(i)}) = g(x) + f(x) = g(x) + f_{(i)}(x) = \widetilde{P}_{(i)}(x)$ for $x \in \mathcal{B}(x^{(i)}, \epsilon)$. Hence $x^{(i)}$ is an local optimal solution to $\widetilde{P}_{(i)}$, and the lemma is proved. By Lemma 3, $x^{(i)}$ is the unique solution to both $P_{(i)}$ and $\widetilde{P}_{(i)}$.

$\qquad\square$

**Proof of Corollary 1.**

*Proof.* For any $i \geq 1$, by Lemma 3, there exists an $x^{(i)} \in \mathbb{R}^n$ that minimizes $g(x) + f_{(i)}(x)$. Thus we have

$$
\begin{aligned}
d^{(i)} &= g(x^{(i)}) + f_{(i)}(x^{(i)}) \\
&= g(x^{(i)}) + \max_{w \in \mathrm{conv}(\mathcal{V}^{(i-1)})} w^\top x^{(i)} \\
&\geq g(x^{(i)}) + \max_{w \in \mathrm{conv}(\mathcal{A}^{(i-1)})} w^\top x^{(i)} \qquad\qquad \rhd\, \mathcal{A}^{(i-1)} \subseteq \mathcal{V}^{(i-1)} \qquad (7) \\
&> g(x^{(i-1)}) + \max_{w \in \mathrm{conv}(\mathcal{A}^{(i-1)})} w^\top x^{(i-1)} \qquad \rhd\, \text{optimality and uniqueness of } x^{(i-1)} \\
&= d^{(i-1)}.
\end{aligned}
$$

On the other hand, by $\mathcal{V}^{(i-1)} \subseteq \mathrm{vert}(B(F))$, we have

$$
\begin{aligned}
d^{(i)} &= \min_{x \in \mathbb{R}^n} \{ g(x) + \max_{w \in \mathrm{conv}(\mathcal{V}^{(i-1)})} w^\top x \} \\
&\leq \min_{x \in \mathbb{R}^n} \{ g(x) + \max_{w \in B(F)} w^\top x \} \qquad\qquad\qquad\qquad\qquad\qquad (8) \\
&= \min_{x \in \mathbb{R}^n} g(x) + f(x)
\end{aligned}
$$

for all $i \geq 0$. $\qquad\qquad\qquad\qquad\qquad\qquad\qquad\qquad\qquad\qquad\qquad\qquad\qquad\qquad\square$

**Proof of Corollary 2.**

*Proof.* Note that each $\mathcal{V}^{(i)}$ determines a unique $d^{(i)}$. Suppose for contradiction that there exists $i_1 \neq i_2$ but $\mathcal{V}^{(i_1)} = \mathcal{V}^{(i_2)}$, then we will have $d^{(i_1)} = d^{(i_2)}$, which contradicts the fact that $\{d^{(i)}\}$ strictly increases. $\qquad\square$

**Proof of Theorem 2**

*Proof.* Since $\mathrm{vert}(B(F))$ has finitely many vertices, there are only finitely many choices of $\mathcal{V}^{(i)} \subseteq \mathrm{vert}(B(F))$. Thus by Corollary 2, Algorithm 1 terminates within finitely many steps.

Suppose for contradiction that when the algorithm terminates at $i = i_0$, $p^{(i_0)} - d^{(i_0)} > \epsilon \geq 0$. Let $\mathcal{A}^{(i_0)} \triangleq \{w \in \mathcal{V}^{(i_0-1)} : w^\top x^{(i_0)} \triangleq f_{(i_0)}(x^{(i_0)})$ and $v^{(i_0)} \in \mathcal{V}(x^{(i_0)})$. Define $\mathcal{V}^{(i_0)} \triangleq \mathcal{A}^{(i_0)} \cup \{v^{(i_0)}\}$ and $f_{(i_0+1)}(x) = \max\{w^\top x : w \in \mathcal{V}^{(i_0)}\}$, then let $x^{(i_0+1)} = \mathrm{argmin}_{x \in \mathbb{R}^n} g(x) + f_{(i_0+1)}(x)$. By the proof of Corollary 1, we have $d^{(i_0+1)} = g(x^{(i_0+1)}) + f_{(i_0+1)}(x^{(i_0+1)}) > d^{i_0}$, so $\mathcal{V}^{(i_0)}$ is different to any $V^{(i)}$ where $i \leq i_0$, and L-KM should not have terminated at $i = i_0$. Thus L-KM would never terminate when $p^{(i)} - d^{(i)} > \epsilon \geq 0$.

$\qquad\qquad\qquad\qquad\qquad\qquad\qquad\qquad\qquad\qquad\qquad\qquad\qquad\qquad\qquad\qquad\qquad\qquad\square$

# D  Duality (Section 4)

To prove the strong duality between $(P_\mathcal{V})$ and $(D_\mathcal{V})$, we first verify the weak duality:

**Theorem 9** (Weak Duality). *The optimal value of primal problem $(P_\mathcal{V})$ is greater than or equal to the optimal value of the dual problem $(D_\mathcal{V})$.*

*Proof.* We first have

$$
\min_{x \in \mathbb{R}^n} \{ g(x) + \max_{w \in \mathrm{conv}(\mathcal{V})} w^\top x \} = \min_{x \in \mathbb{R}^n} \max_{w \in \mathrm{conv}(\mathcal{V})} g(x) + w^\top x. \qquad (9)
$$

For any given $\tilde{w} \in \mathrm{conv}(\mathcal{V})$, we also have

$$
\min_{x \in \mathbb{R}^n} \max_{w \in \mathrm{conv}(\mathcal{V})} g(x) + \tilde{w}^\top x \geq \min_{x \in \mathbb{R}^n} g(x) + \tilde{w}^\top x. \qquad (10)
$$

Thus by the definition of $g^*$, we can see that

$$
\begin{aligned}
\min_{x \in \mathbb{R}^n} \max_{w \in \mathrm{conv}(\mathcal{V})} g(x) + w^\top x &\geq \max_{w \in \mathrm{conv}(\mathcal{V})} \min_{x \in \mathbb{R}^n} g(x) + w^\top x \\
&= \max_{w \in \mathrm{conv}(\mathcal{V})} - \max_{x \in \mathbb{R}^n} (-w)^\top x - g(x) \qquad (11) \\
&= \max_{w \in \mathrm{conv}(\mathcal{V})} -g^*(-w).
\end{aligned}
$$

Combine (9) and (11), and the theorem follows. $\qquad\qquad\qquad\qquad\qquad\qquad\qquad\qquad\qquad\square$

**Proof of Theorem 3**

*Proof.* By Lemma 3, we know ($P_{\mathcal{V}}$) has a unique solution $\bar{x}$. Since $g$ is convex, we have $\partial g(x) \neq \emptyset$. By the optimality of $\bar{x}$, we also have $0 \in \partial g(\bar{x}) + \partial f(\bar{x})$. Let $\bar{w} \in -\partial g(\bar{x}) \cap \partial f(\bar{x})$, then

$$g^*(-\bar{w}) = (-\bar{w})^\top \bar{x} - g(\bar{x}) \tag{12}$$

by Eq. (4). Note that $\bar{w} \in \partial f(\bar{x})$, we also have $f(\bar{x}) = \bar{w}^\top \bar{x}$ by Equation (2). Thus

$$f(\bar{x}) + g(\bar{x}) = g^*(\bar{w}), \tag{13}$$

$\bar{w}$ is an optimal solution to ($D_{\mathcal{V}}$) and we have ($P_{\mathcal{V}}$) and ($D_{\mathcal{V}}$) via weak duality. $\qquad\square$

# E   Primal-from-dual algorithm (Section 5)

Now consider the *Primal-from-dual* algorithm presented in Section 5.

Formally, assume $g$ is $\alpha$-strongly convex. Suppose we obtain $w \in B(F)$ with

$$\|w - w^\star\| \leq \epsilon$$

via some dual algorithm (e.g., L-FCFW). Define $x = \nabla_w(-g^*(-w)) = \operatorname{argmin}_x g(x) + w^\top x$. Since $g^*$ is $1/\alpha$ smooth, we have

$$\|x - x^\star\| \leq 1/\alpha \|w - w^\star\| \leq \epsilon/\alpha$$

Hence if the dual iterates converge linearly, so do the primal iterates.

The remaining difficulty is how to solve the L-FCFW subproblems. One possibility is to use the values and gradients of (a FIRST ORDER ORACLE for) $h = g^*$. To implement a first order oracle for $h = g^*$, we need only solve an unconstrained minimization problem:

$$g^*(y) = \max_{x \in \mathbb{R}^n} y^\top x - g(x), \qquad \nabla g^*(y) = \operatorname*{argmax}_{x \in \mathbb{R}^n} y^\top x - g(x).$$

This problem is straightforward to solve since $g$ is smooth and strongly convex. However, it is not clear how solving these subproblems approximately affects the convergence of L-FCFW. Morever, we will see in the next section that L-KM achieves exactly the same sequence of iterates as the above (rather unwieldly) proposal.

# F   Duality between L-KM and L-FCFW (Section 6)

**Lemma 7.** *Only vertices in $\mathcal{A}^{(i)}$ can have positive convex multipliers in the convex decomposition of $w^{(i)}$, i.e., if we write $w^{(i)} = \sum_{v \in \mathcal{V}^{(i-1)}} \lambda_v^{(i)} v$ such that $0 \leq \lambda_v \leq 1$ for any $v \in \mathcal{V}^{(i-1)}$, then $\lambda_v^{(i)} = 0$ for any $v \in \mathcal{V}^{(i-1)} \setminus \mathcal{A}^{(i)}$.*

*Proof.* By the definition of $\mathcal{A}^{(i)}$, we have

$$\begin{aligned}
\operatorname{conv}(\mathcal{A}^{(i)}) &= \operatorname{conv}(\{v \in \mathcal{V}^{(i-1)} \mid v^\top x^{(i)} = {w^{(i)}}^\top x^{(i)}\}) \\
&= \operatorname{conv}(\{v \in \mathcal{V}^{(i-1)} \mid v^\top x^{(i)} = \max_{w \in \operatorname{conv}(\mathcal{V}^{(i-1)})} w^\top x^{(i)}\}) \\
&= \arg\max_{w \in \operatorname{conv}(\mathcal{V}^{(i-1)})} w^\top x^{(i)}.
\end{aligned} \tag{14}$$

Then

$$\begin{aligned}
0 &= (w^{(i)} - w^{(i)})^\top x^{(i)} \\
&= (w^{(i)} - \sum_{v \in \mathcal{V}^{(i-1)}} \lambda_v^{(i)} v) \\
&= \sum_{v \in \mathcal{V}^{(i-1)} \setminus \mathcal{A}^i} \lambda_v^{(i)} [(w^{(i)})^\top x^{(i)} - v^\top x^{(i)}]. \qquad \triangleright\, v^\top x^{(i)} = {w^{(i)}}^\top x^{(i)}, \forall\, v \in \mathcal{A}^{(i)}
\end{aligned} \tag{15}$$

Using (14), we have $v^\top x^{(i)} - w^{(i)} x^{(i)} < 0$ for any $v \in \mathcal{V}^{(i-1)} \setminus \mathcal{A}^{(i)}$. Thus $\lambda_v^{(i)} = 0$ for any $v \in \mathcal{V}^{(i-1)} \setminus \mathcal{A}^{(i)}$. $\qquad\square$

**Proof of Theorem 6.**

*Proof.* We prove by induction. When $i = 1$, $\mathcal{V}^{(0)}$ will naturally refer to the same set of points in L-KM and L-FCFW. By Lemma 3, we have $x^{(1)}$ is the unique solution to $g + f_{(1)}$. Note $g^*$ is strongly convex given $g$ is smooth (Lemma 5), we have $w^{(1)}$ is the unique solution to $\max_{w \in \text{conv}(\mathcal{V}^{(0)})} -g^*(-w)$. Let $\mathcal{V} = \mathcal{V}^{(0)}$ in Theorem 4, we have that $x^{(1)} = -\nabla g^*(-w^{(1)})$ is the unique minimizer of $g + f_{(1)}$. So $x^{(1)}$ in the two algorithms match. Also note that $w^{(1)}$ solves $\max_{w \in \text{conv}(\mathcal{V}^{(0)})} -g^*(-w)$, we have $w^{(1)}$ maximizes $w^\top x^{(1)}$ for all $w \in \text{conv}(\mathcal{V}^{(i-1)})$ by the first order optimality condition, which gives $w^{(i)\top} x^{(i)} = f_{(i)}(x^{(i)})$. Thus $\mathcal{A}^{(1)}$, $\mathcal{V}^{(1)}$ match consequently. By strong duality in Theorem 3, we have $d^{(1)}$ matches in the two algorithms. Note $g^*$ is strongly convex, which gives the uniqueness of $w^{(1)}$. By Theorem 4, $\nabla g(w^{(1)})$ solves the primal subproblem, so $x^{(1)}$ match in the two algorithms by the uniqueness of $x^{(1)}$.

Suppose that the theorem holds for $i = i_0$, in particular, the $\mathcal{V}^{(i_0)}$ match in the two algorithms. Then for $i = i_0 + 1$, we can use the same argument as in the previous paragraph by substituting 0 with $i_0$ and 1 with $i_0 + 1$, and show that all the statements hold for $i = i_0 + 1$. Note that by Lemma 7, $\mathcal{A}^{(i)}$ satisfies the condition in Line 6 of L-FCFW. Thus this theorem is valid. $\qquad\square$

# G  Duality between OSM and L-FCFW (Section 6)

**Theorem 10.** *If $g$ is smooth and strongly convex and in Algorithm 2 we choose $\mathcal{B}^{(i)} = \mathcal{V}^{(i-1)}$, then*

1. *The primal iterates $x^{(i)}$ of Algorithm 3 and Algorithm 2 match.*

2. *The set $\mathcal{V}^{(i)}$ used at each iteration of Algorithm 3 and Algorithm 2 match.*

3. *The upper and lower bounds $p^{(i)}$ and $d^{(i)}$ of Algorithm 3 and Algorithm 2 match.*

The proof of Theorem 10 is similar to the proof of Theorem 6.

# H  Definition of Diameter and Pyramid Width

**Diameter.** The diameter of a set $\mathcal{P} \subseteq \mathbb{R}^n$ is defined as

$$\text{Diam}(\mathcal{P}) \triangleq \max_{v,\,w \in \mathcal{P}} \|v - w\|_2. \tag{16}$$

**Directional Width.** Given a direction $x \in \mathbb{R}^n$, the directional width of a set $\mathcal{P} \subseteq \mathbb{R}^n$ with respect to $x$ is defined as

$$\text{dirW}(\mathcal{P},\,x) \triangleq \max_{v,\,w \in \mathcal{P}} (v - w)^\top \frac{x}{\|x\|_2}. \tag{17}$$

Pyramid directional width and pyramid width are defined by Lacoste-Julien and Jaggi in [15] for a finite sets of vectors $\mathcal{V} \subseteq \mathbb{R}^n$. Here we extend the definition of pyramid width to a polytope $\mathcal{P} = \text{conv}(V)$, and it should be easy to see that the two definitions are essentially the same.

**Pyramid Directional Width.** Let $\mathcal{V} \subseteq \mathbb{R}^n$ be a finite set of vectors in $\mathbb{R}^n$. The pyramid directional width of $\mathcal{V}$ with respect to a direction $x$ and a base point $w \in \text{conv}(\mathcal{V})$ is defined as

$$\text{PdirW}(\mathcal{V},\,x,\,w) \triangleq \min_{A \in \mathcal{A}(w)} \text{dirW}(A \cup \{v(\mathcal{V},\,x)\},\,x), \tag{18}$$

where $\mathcal{A}(w) \triangleq \{A \subseteq \mathcal{V} \mid \text{the convex multipliers are non-zero for all } v \in A \text{ in the decomposition of } w\}$ and $v(\mathcal{V},\,x)$ is a vector in $\arg\max_{v \in \mathcal{V}} v^\top x$. The pyramid directional width got its name because the set $A \cup \{v(\mathcal{V},\,x)\}$ has the shape of a pyramid with $A$ being the base and $v(\mathcal{V},\,x)$ being the summit.

**Pyramid Width.** The pyramid width of $\mathcal{P}$ is defined as

$$\text{PWidth}(\mathcal{P}) \triangleq \min_{\mathcal{K} \in \text{face}(\mathcal{P})} \min_{x \in \text{cone}(\mathcal{K}-w)\setminus\{0\},\, w \in \mathcal{K}} \text{PdirW}(\mathcal{K} \cap \text{vert}(\mathcal{P}),\,x,\,w), \tag{19}$$

where $\text{face}(\mathcal{P})$ stands for the faces of $\mathcal{P}$ and $\text{cone}(\mathcal{K} - w)$ is equivalent to the set of vectors pointing inwards $\mathcal{K}$.