[Reviews · NeurIPS 2018]

Reviewer 1



To optimize a composite objective of a convex function and a submodular function, the paper proposes a limited memory Kelley's method with a convergence rate analysis. The method is a variant of OSM with no rate of convergence known. The paper also connects their method with FCFW and show comparable performance with synthetic experiments. I'm not very familiar with the Kelley's method or OSM, but based on what I read, the paper does consider interesting applications and settings. The method proposed with the theoretical analysis is clear. The analysis of the connection between DL-KM and FCFW is enlightening. The experimental results are apt and encouraging. I want to propose some issues that the paper could improve upon. The introduction is not very well-structured. It renders from Kelley's method, the definition of sub modularity, to the problem formulation, and OSM's advantages and drawbacks, which followed by related work part again. Logic doesn't flow very well. The second paragraph of the application part compares the proposed method with MD and looks more like background of the problem. The summary of contribution is actually just the outline of the remaining paper. Most of the cited work is very old. The paper should add some newest reference about bundle methods and FCFW, e.g. [1,2,3], to name a few. [1] W. de Oliveira and C. Sagastiz´abal. Bundle methods in the XXIst century: A birds-eye view. Optimization Online Report, 4088, 2013 [2] Jaggi, Martin. "Revisiting Frank-Wolfe: Projection-Free Sparse Convex Optimization." ICML (1). 2013. [3] Lacoste-Julien, Simon, and Martin Jaggi. "On the global linear convergence of Frank-Wolfe optimization variants." Advances in Neural Information Processing Systems. 2015. In the intro, the authors mentioned that DL-KM is favorable than MD since Bregman projections is more computational complex. But have you done such analysis later? What is the per iteration complexity of the proposed methods?


Reviewer 2



The paper proposes a limited memory version of Kelly's cutting plane method for minimization of composite functions of the form g(x) + f(x) where g is closed convex and importantly f is the Lovasz extension of a submodular function. The authors build upon Original Simplical Method (OSM) where one approximates f iteratively by f_{(i)}, the maximum of some supporting hyperplanes of f and f_{(i)} gets closer to f with each iteration, as a new bunch of hyperplanes are added. In the proposed method, in iteration i+1, they keep only those hyperplanes that touch f_{(i+1)} from the previous iteration i, whereas in OSM, all the previous ones are carried forward. They show that the number of hyperplanes will at most be n+1 where x \in R^n. They show convergence rate of the method via the relation between its dual and the Fully-Corrected Frank-Wolfe with approximations. This is a key contribution of the paper. Simulations are supportive but not too extensive. The authors organize the paper well, explain their work in a clear manner at most places and relate it to previous work. The method is novel and the convergence result is very interesting.

Reviewer 3



Summary: ------- The paper proposes a simple modification of the Original Simplicial Method that keeps in memory only the active linear constraints. Then by linking the dual method to Fully corrective Frank Wolfe they get convergence rates. Quality & Clarity: ------------------ The paper is well written and motivated for people in the field. Note that it is probably quite abstruse otherwise. Its place in the literature is well defined. Proofs are correct. The final rates (Theorem 2 and 3) are quickly stated: what is the total cost of the algorithm, taking into account the computations of the subproblems ? (by taking a concrete example where the subproblems can be computed efficiently) This would enlighten the impact in this rate of the limited memory implementation in comparison to the original simplicial method. In Theorem 3, first, \Delta^{(i+1)} should be \Delta^{(i)}. Then these results only state that the estimated gap is bounded by the true gap, I don't get the sub-linear rate from reading these equations. Originality & Significance: --------------------------- The paper introduces two simple, though efficient, ideas to develop and analyze an alternative of the simplicial method (looking only at the active linear constraints of each subproblem and drawing convergence rates from the Frank-Wolfe literature). Relating the dual of the simplicial method to Franck-Wolfe method was already done by Bach [2]. I think that the present paper is therefore only incremental. Moreover the authors do not give compelling applications (this is due to the restricted application of the original simplicial method). In particular, they do not give concrete examples where the subproblems can easily be computed, that would help to understand the practical impact of the method. Minor comments: --------------- - l139: no parenthesis around w - l140: put footnote for closed function here - l 227: Replace "Moreover" by "Therefore" - l 378: must be strictly contained *in* - As stated from the beginning, OSM is not Kelley's method, so why is the algorithm called Limited memory Kelley's method and Limited memory simplicial method ? This confusion might weaken its referencing in the literature, I think. -------------------------------------- Answer to author's feedback -------------------------------------- Thank you for your detailed feedback. I revised my "incremental" comment. The problem was clearly asked before and the answer you provided is simple, clear and interesting.